# Precision Layered Stealth Dicing of SiC Wafers by Ultrafast Lasers

**DOI:** 10.3390/mi13071011

**Published:** 2022-06-27

**Authors:** Bo Yang, Heng Wang, Sheng Peng, Qiang Cao

**Affiliations:** The Institute of Technological Sciences, Wuhan University, Wuhan 430072, China; yangb15@whu.edu.cn (B.Y.); wangheng2021@whu.edu.cn (H.W.); pengsheng@whu.edu.cn (S.P.)

**Keywords:** ultrafast laser, stealth dicing, silicon carbide (SiC), wafer dicing, precision layered stealth dicing

## Abstract

With the intrinsic material advantages, silicon carbide (SiC) power devices can operate at high voltage, high switching frequency, and high temperature. However, for SiC wafers with high hardness (Mohs hardness of 9.5), the diamond blade dicing suffers from problems such as debris contaminants and unnecessary thermal damage. In this work, a precision layered stealth dicing (PLSD) method by ultrafast lasers is proposed to separate the semi-insulated 4H-SiC wafer with a thickness of 508 μm. The laser power attenuates linearly from 100% to 62% in a gradient of 2% layer by layer from the bottom to the top of the wafer. A cross section with a roughness of about 1 μm was successfully achieved. We have analyzed the effects of laser pulse energy, pulse width, and crystal orientation of the SiC wafer. The anisotropy of the SiC wafer results in various qualities of PLSD cross sections, with the roughness of the crystal plane {10−10} being 20% lower than that of the crystal plane {11−20}.

## 1. Introduction

On the occasions of high voltage, high switching frequency, and high temperature, the wide-bandgap semiconductor silicon carbide (SiC), with the advantages of high efficiency and severe-environment adaptability, has gradually replaced silicon devices, widely applied to energy-efficient lighting, fifth-generation (5G) mobile communications, electric vehicles, smart grid, rail transit, new energy, intelligent manufacturing, and radar detection [1,2,3,4]. One of the essential processes in integrated circuit (IC) chip manufacturing is wafer dicing, which splits the wafer into thousands of tiny chips. The dicing quality has a great impact on wafer utilization and IC chip quality.

For SiC wafers with a Mohs hardness of 9.5, there are many problems when using the diamond blade dicing method, such as poor cutting quality, low efficiency, and low wafer utilization [5]. In diamond blade dicing, a diamond blade is used to carve kerf marks on the wafer surface, and the wafer is split along the kerf mark. However, when applied to hard SiC wafers, it is likely to produce mechanical stress and lead to kerf edge breakage and surface damage. Jinn P. chu et al. [6] researched the coating of diamond dicing blades with a Zr-based metallic glass coating to improve the kerf quality with a low coefficient of friction, but the kerf width was still around 50 µm. Diamond blade dicing has some inherent disadvantages that are difficult to overcome completely. A new method of wafer dicing with high quality is valuable to the SiC semiconductor industry.

Compared to diamond blade dicing, non-contact laser dicing can solve problems caused by tool wear and mechanical stress, achieving high quality and precision of the kerfs. There are various laser-based methods such as thermal laser separation, laser ablation, and stealth dicing (SD). Thermal laser separation (TLS) is a novel dicing technology for large-scale production of SiC devices [7]. With feed rates up to two orders of magnitude higher than mechanical blade dicing, no tool wear, and high quality of diced chips, TLS is a potential technology for SiC wafer dicing.

Pal Molian et al. [8] used an ultra-short, pulsed laser with a 1552 nm wavelength, 2 ps pulse, and 5 mJ pulse energy ablating on an n-type 4H-SiC wafer to achieve rapid etching of diaphragms for pressure sensors. Trenches of 26 mm deep and holes of 10 mm diameter on the SiC wafer were machined with clean edges and a smooth bottom surface. It was concluded that picosecond pulsed laser ablation is more effective than a femtosecond pulsed laser in terms of ablation rate and nanosecond pulsed laser in terms of cleanliness and thermal effects. However, the thermal damage on the surface still exists, and the depth of ablation is limited to about 150 µm [9].

SD focuses the laser inside the material to form a modified layer and then separates the chips by means of tape expansion or other external tensile force [10]. Kumagai M. et al. [11] applied a nanosecond laser to the silicon wafer dicing process, proving that the method has the advantages of no debris contaminants, narrow kerfs, and high processing efficiency. Ohmura E. et al. [12] focused a permeable nanosecond laser beam inside a silicon wafer and induced a high dislocation density layer after the thermal shock wave propagation. The possibility of internal crack propagation by laser pulses was investigated, which indicated that the internal cracks were already generated before division. Compared to nanosecond lasers, ultrafast lasers induce quite special phenomena, such as nonlinear multi-photon absorption, extreme non-equilibrium between electrons and lattices, non-thermal phase transition, coulomb explosion, and electrostatic ablation [13,14,15]. The ultrafast laser has shown numerous advantages in the field of micro/nano-fabrication since its advent [16,17]. Lee et al. [18] and Zhang et al. [19] have compared the SD effect of an ultrafast laser and nanosecond laser on transparent materials and concluded that an ultrafast laser SD can reduce the debris and thermal damage of kerfs compared with a nanosecond laser. Kim E. et al. [20] sliced off a thin wafer from a bulk SiC crystal along the crystal plane {0001} by using femtosecond laser double pulses with an exfoliated surface roughness of 5 μm and a cutting-loss thickness smaller than 24 μm. Amit Yadav et al. [21] have investigated the SD of sapphire wafers with near-infrared femtosecond pulses to leave a high-quality surface, and Caterina Gaudiuso et al. [22] applied a femtosecond laser to SD on quartz glass. Zhang et al. [23] proposed a new asynchronous double-beam laser SD method, in which an ultrafast laser is focused inside of the SiC wafer along the writing line to form microcracks, heated by continuous lasering to expand the cracks through thermal stress, achieving the dicing of a 200 μm SiC wafer with almost no damage to the kerf. However, it is only suitable for thicknesses less than 200 µm.

The trend of IC chip integration promotes the continuous reduction of chip size, and the advantages of ultrafast laser micro-nanomachining will be further prominent on thin wafer dicing. Ultrafast laser SD has a very narrow width of the kerf, contributing to the tiny width of the reserved dicing line on the wafer, and the chips on the same wafer can be arranged more densely, thus increasing the number of chips on a single wafer and improving the utilization rate of the wafer. The realization of SD on SiC wafers with high quality and high efficiency is expected to break through the technical bottleneck and the limit of high cost and promote the market application of SiC devices in place of silicon substrates.

In this work, based on the physical properties of SiC material, we have proposed a precision layered stealth dicing (PLSD) method by ultrafast lasers to successfully separate the semi-insulated 4H-SiC wafer with a thickness of 508 μm. It is compact and suitable for industrial applications. We believe that the method can be applied to dicing SiC wafers and has a promising potential to fulfill the demands of the IC manufacturing of 4H-SiC devices. The effects of laser pulse energy, pulse width, and crystal orientation of the SiC wafer on the quality of kerfs were analyzed.

## 2. Experiments

The experimental object of ultrafast laser PLSD was a commercially available semi-insulated 4H-SiC single crystal wafer (Tankeblue semiconductor) with a thickness of 508 μm. An ultrafast laser source (PHAROS, SP-HP) was used to investigate PLSD of the semi-insulated 4H-SiC semiconductor wafer. The ultrafast laser, which has pulse width of 5 ps, repetition frequency of 200 kHz, and laser radius of 1.13 μm, was tightly focused inside the SiC wafer through a 100× objective lens with numerical aperture (NA) of 0.7, and the incident surface was the (0001) silicon plane of SiC wafer, as shown in Figure 1a. The pulse energy was 45 μJ with laser power of 9 W. With the coordination of the XYZ 3 axis precision moving platform (Aerotech, XYZ accuracy ± 0.3 µm), the laser focus is moved along the preset path with a scanning speed of 10 mm/s. The scan direction was parallel or perpendicular to the primary flat orientation of the wafer, that is, along the crystal direction <11−20>. The laser pulse spots overlapped along the scanning direction with high density, and the dense bonding of volume elements created an SD modified layer running through both edges of the wafer. The internal stress and microcracks were formed in the modified layer, which provided the basis for subsequent wafer separation.

After laser scanning, a modified layer was derived inside the wafer along the scanning direction. However, the single SD layer was insufficient for chip separations of the hard SiC wafer with a thickness of 508 μm. Therefore, the PLSD method was designed, as shown in Figure 1b. An ultrafast laser was applied to scan inside SiC wafer from bottom to top, with the laser focus moving up after each layer, to achieve multiple layers of machining and a modified section. Considering the wafer thickness of 508 μm, the PLSD layer vertical spacing in SiC wafer was set to 20 μm, divided into 24 layers. However, there is a problem that ablation may occur when the laser is focused closely to the surface of the wafer. To avoid ablations on the top or the bottom of the wafer after laser penetration, 3 layers near the top and 1 layer near the bottom were not processed, as the number of actual scanning layers was 20. Furthermore, to compensate for the difference of transmitted laser intensity at different depths from the top surface and equate the energy deposition of each layer, the laser power attenuated linearly from 100% to 62% in a gradient of 2% layer by layer from the bottom to the top by a motorized attenuator. Dense microcracks existed in the layers after PLSD, and the chip separation of semi-insulated 4H-SiC wafer was achieved by applying tensile force to the PLSD section of the wafer, as shown in Figure 1c.

One of the essential foundations for PLSD is that the wafer is transparent to the wavelength of the laser used. The 4H-SiC wafer sample was characterized by absorption and transmission spectra from ultraviolet to infrared wavebands, using ultraviolet and visible spectrophotometer (UV-2600i). Figure 1d showed the spectra characterization in the range of 300~1200 nm of semi-insulated 4H-SiC wafer. It is indicated that the transmittance of 4H-SiC wafer is higher than 60% in the wavelength band larger than 400 nm. In experiments, the laser center wavelength was 1028 nm, with a corresponding transmittance of 65.8%. The minimum laser power attenuation of 62% is close to that.

The morphology of the kerfs was tested using scanning electron microscopy (SEM, TESCAN, Brno, Czech, MIRA 3), and their cross sections were measured by a 3D optical surface profilometer (ZYGO, Middlefield, CT, USA, New View 9000).

## 3. Results and Discussion

### 3.1. Effect of Laser Pulse Energy

Laser average power and repetition frequency collectively determine the pulse energy of ultrafast lasers. The pulse energy $E_{0}$ has a relationship with laser power P and repetition frequency f as:(1)$$E_{0}=\frac{P}{f}$$

Figure 2 showed the ablation effects of the SiC wafer surface with various pulse energies. The laser focus was located at 80 μm below the surface, belonging to shallow layer experiments with a laser power of 9 W. It can be seen that when the laser pulse energy was 45 μJ, there was only a tiny heat-affected region at the surface with a line width of approximately 8 μm, as shown in Figure 2a. The overall effect on the wafer surface seemed not obvious. Figure 2b showed that in the case of 90 μJ, the surface was slightly ablated, and the line width appeared to be about 10 μm. As pulse energy increased to 180 μJ, severe ablation occurred at the surface, and the line width reached 33 μm, causing severe ablation damage to the wafer kerf, as shown in Figure 2c. It was indicated that low pulse energy can reduce the energy deposition on the wafer surface without ablation of the wafer surface, approaching real “stealth”.

However, this did not mean that the higher the pulse energy the better. With the upper limit of laser power in the ultrafast laser machining system, of which the maximum power could be adjusted to 10 W, the low pulse energy caused the lack of energy deposition to induce modified layers, and effective PLSD could not be realized in the wafer. Therefore, in the specific PLSD experiments, it was necessary to select the laser power and repetition frequency properly according to the wafer material properties, such as transmittance and ablation threshold. For the semi-insulated 4H-SiC wafer, the relatively proper pulse energy was 45 μJ under our experimental conditions.

### 3.2. Effect of Laser Pulse Width

Das A et al. [24] have defined the threshold conditions of initiating modifications inside silicon by the pulse duration and the numerical aperture, with a modest pulse duration dependence of the energy threshold. In order to investigate the influence of laser pulse width on the effects of PLSD, a series of experiments with different pulse widths from femtosecond to picosecond were carried out. With a pulse width of hundreds of femtoseconds, a modification line with several microns was formed inside the SiC wafer. However, the size of the internal modification region was not large enough to separate the wafer by tensile force. When the pulse width reached sub-picosecond and picosecond, wafer separation was accomplished successfully.

Figure 3 shows the images of the morphology of cross sections with different pulse widths after wafer separation. It can be seen that the crack size seemed smaller, and the crack distribution was denser with the pulse width of 1 ps, compared to other cases. When the pulse width modulation was amplified to 5 ps or 10 ps, the crack size increased gradually. The quality of the cross section was accurately measured by the size of the surface roughness. The surface average roughness (Sa) of the cross sections was obtained by a 3D optical surface profilometer, with a scanning area size of 350 × 350 μm^2^. When the pulse width was 1 ps, the Sa of the cross section showed 0.699 μm. When the pulse width reached 5 ps, the Sa increased to 1.613 μm. When approaching 10 ps, the Sa increases to 1.649 μm. It is obvious that the roughness of the cross section rose as the pulse width increased, and the increase in roughness was not significant for larger pulse widths. The crack size of the cross section gradually increased as the laser pulse width increased. When the pulse width was small, the crack size was small and densely distributed, and the trajectory of PLSD was covered by microcracks. While the pulse width was large, PLSD layers along the scanning direction could be clearly seen, and there are also some cracks and fractures extending inward and outward through the adjacent PLSD layers.

In general, the laser pulse width has a great influence on the PLSD results. Femtosecond lasers can achieve internal modification of SiC wafers, forming modified regions of only a few microns to a dozen microns in width, and wafer separation is hard to achieve. Picosecond lasers induce a modified region large enough to achieve the separation of SiC wafers. The larger the pulse width, the larger the crack size of the cross section, the higher the roughness, and the greater the degree of damage inside the wafer, which means that the wafer is more likely to separate. In order to obtain the best quality of PLSD, the appropriate laser pulse width needs to induce a large enough modified region inside the wafer to achieve wafer separation while minimizing the roughness of the cross section. The optimal pulse width is varied for different materials and may be generally at picosecond and sub-picosecond levels.

### 3.3. Effect of Crystal Orientation

For hexagonal SiC single crystals, different crystal orientations mean anisotropic mechanical properties [25], such as fracture toughness values, so different orientations of PLSD can result in various kerf morphologies. Figure 4 shows the SEM images of the kerfs after ultrafast laser PLSD of SiC wafers obtained under a 45° oblique view. From Figure 4b,c, it can be seen that the kerf edges along the crystal orientations <11−20> and <10−10> were both in good agreement within a length of 1.7 mm along the dicing line. Some minor notches existed in the local section, and the straightness was within 20 μm. However, there was a significant distinction on the morphology of the cross sections of the two crystal planes {10−10} and {11−20}. The cross section of crystal plane {10−10} was smoother than that of crystal plane {11−20}, indicating the influence of the anisotropy of the crystal orientations or crystal planes on the kerf quality of PLSD.

The cross sections of the PLSD kerfs of Figure 4 along different crystal orientations or planes were obtained by a 3D optical surface profiler, as shown in Figure 5a,b. On the cross sections of PLSD kerfs, the different cleavage plane exhibited a large difference in roughness. The profiler scanned two cross sections of crystal planes, {10−10} and {11−20}, with an area of 434 × 434 μm^2^. The surface roughness values were 0.894 μm and 1.126 μm, respectively, which meant that the surface roughness of crystal plane {10−10} was 20% lower than that of crystal plane {11−20}. This indicated that plane {10−10} is easier to cleave than plane {11−20}. This might be related to crystal cleavability, which is the capacity of the crystal cleavage along one specific crystal plane under the action of an external force. The cleaved crystal plane was called a cleavage plane, which was generally a crystal plane with weak bonding of atomic layers. The material properties [26] of 4H-SiC allow the calculation of theoretical fracture toughness values [27] for both crystal planes:(2)$${\left(K_{IC}\right)}_{hkl}={\left(E_{hkl}{\left(G_{IC}\right)}_{hkl}\right)}^{1/2}\;MPa\;m^{1/2}$$

Here, $E_{hkl}$ is the elastic modulus and ${\left(G_{IC}\right)}_{hkl}$ is the critical crack energy release rate. The fracture toughness of 1.4 MPa m^1/2^ was theoretically calculated for the primary cleavage plane {10−10}, and 1.8 MPa m^1/2^ was calculated for the secondary cleavage plane {11−20} [28]. Compared to {11−20}, the fracture toughness of {10−10} is 22% lower, which is similar to the difference of 20% in the roughness of two cross sections. This explains why the kerf quality of {10−10} was better than that of {11−20} under the same experimental conditions.

In addition, for the crystal planes of the hexagonal crystal system (hkil), the crystal plane spacing d(hkl) can be calculated from the lattice constants as:(3)$$d\left(hkl\right)=\frac{1}{\sqrt{\frac{3}{4}\left(\frac{h^{2}+hk+k^{2}}{a^{2}}\right)+{\left(\frac{l}{c}\right)}^{2}}}$$

The crystal plane spacings of 4H-SiC were calculated to be 3.55 Å and 2.05 Å for crystal planes {10−10} and {11−20}, respectively. The larger crystal plane spacing of {10−10} means weaker bonding of the atomic layers and better crystal cleavability, which agrees with the better quality of {10−10} than {11−20}.

The comparison of Figure 6b,c showed that the microcrack size inside the modified region was smaller and the roughness was lower, while the roughness was relatively higher near the top and the bottom surfaces. This was due to the fact that the PLSD layers were mainly inside the wafer, and in order to reduce the thermal effect on the wafer surfaces, there was no laser modification near the top and bottom surfaces. When the wafer was separated by tensile force after PLSD, the stress inside the modified region was released to the top and bottom surfaces as the cracks extended outwards. The roughness was relatively higher because of the lack of guidance for the PLSD layers near the top and bottom surfaces. This is a problem that needs to be solved in further studies to avoid surface thermal effects while reducing the overall roughness of the kerf edge and improving the kerf quality.

## 4. Conclusions

We have proposed a precision layered stealth dicing method by ultrafast lasers to separate SiC wafers with high-quality kerfs. A cross section with a roughness of about 1 μm was successfully achieved. In addition, the effects of the significant parameters in PLSD processing on the kerf quality were experimentally investigated. It was showed that low pulse energy can reduce the damage at the wafer’s surface. It is more difficult for the femtosecond laser to separate the SiC wafer after PLSD than the picosecond laser, due to the insufficient size of the internal modified region. A shorter pulse width corresponds to a smaller crack size and lower roughness of the cross sections. Anisotropic crystal orientations have different crystal cleavability and fracture toughness values. The cross section along the primary cleavage plane {10−10} is visibly smoother than the secondary cleavage plane {11−20}. The surface roughness of the crystal plane {10−10} is 20% lower than that of {11−20}, which corresponds to a 22% lower fracture toughness. The PLSD by ultrafast lasers is a non-contact dry process with no debris generation and almost no effect on the wafer surface, which has great potential in the dicing of hard-brittle materials and IC chip manufacturing.

## Figures and Tables

**Figure 1 micromachines-13-01011-f001:**
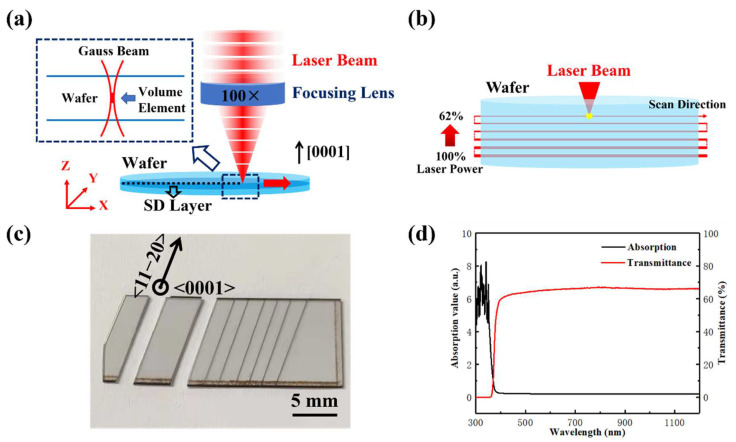
(**a**) Schematic diagram of ultrafast laser scanning in the SiC wafer and (**b**) PLSD. (**c**) The semi-insulated 4H-SiC wafer sample after PLSD. (**d**) Absorption and transmission spectra of the wafer at 300–1200 nm.

**Figure 2 micromachines-13-01011-f002:**
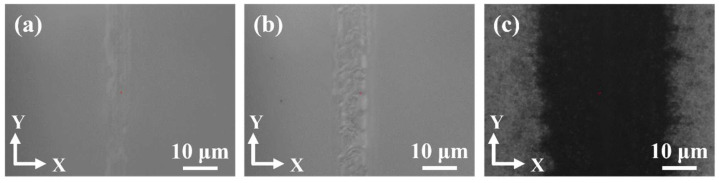
The optical images of SiC wafer surface with the various pulse energies of (**a**) 45 μJ, (**b**) 90 μJ, and (**c**) 180 μJ.

**Figure 3 micromachines-13-01011-f003:**
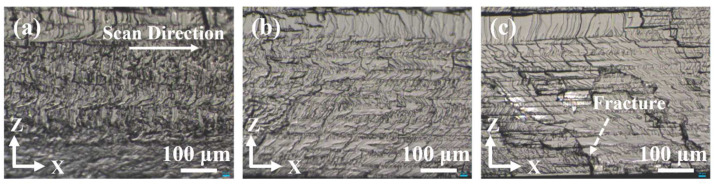
Morphology of cross sections with different pulse widths of (**a**) 1 ps, (**b**) 5 ps, and (**c**) 10 ps.

**Figure 4 micromachines-13-01011-f004:**
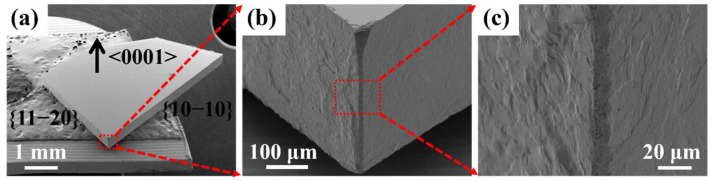
SEM images of the kerfs after ultrafast laser PLSD of SiC wafers with the magnification of (**a**) 40×, (**b**) 500× and (**c**) 2000×.

**Figure 5 micromachines-13-01011-f005:**
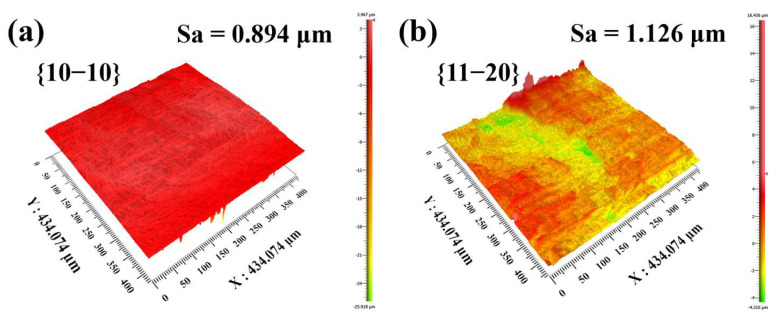
Cross sections of the PLSD kerfs along different crystal orientations or planes. (**a**) The cross section of crystal plane {10−10} and (**b**) the cross section of crystal plane {11−20}.

**Figure 6 micromachines-13-01011-f006:**
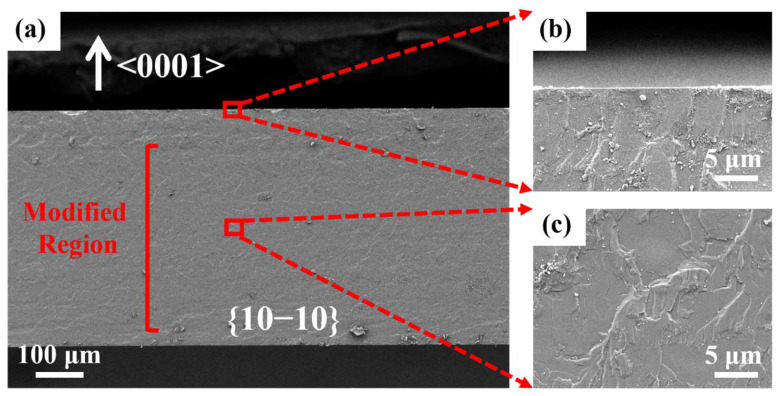
SEM images of (**a**) cross section of the SiC wafer after PLSD, (**b**) enlarged view near the top surface, and (**c**) enlarged view at the center of the modified region.

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
