# Peer review of "Precision Layered Stealth Dicing of SiC Wafers by Ultrafast Lasers"

_micromachines, 2022, doi:10.3390/mi13071011_

Round 1

Reviewer 1 Report

Comment-1: In introduction, the motivation for the present research would be clearer if the author could provide a more direct link between the importance of choosing an appropriate parameters and more references from other groups who have done research in this area should be cited especially within 5 years.

Comment-2: The author should further explain the method and its novelty for this present work in more details.  

Comment-3: PLSD is suggested to be included in the keywords given that it was used in multiple occasions.

Comment-4: Figure 1(b) could be described better.

Comment-5: This paper is comprehensive and you managed to successfully discuss the importance of your research, from both a theoretical and an applied perspective.

Comment-6: The paper should be re-edited by native English speaker. There are some mistakes about the Figures and references.

Reviewer 2 Report

The following literature has used stealth dicing (SD) to conduct similar studies, and this paper does not specify the difference from [9]

[9] Kim, E.; Shimotsuma, Y.; Sakakura, M.; Miura, K. 4H-SiC wafer slicing by using femtosecond laser double-pulses. Optical Ma- 275

terials Express 2017, 7, doi:10.1364/ome.7.002450.

Reviewer 3 Report

1. Typical references on the femtosecond laser microfabrication of SiC crystals such as slicing, modification, and ablation need to be reviewed and analyzed.

2. Does the polarization state of the ultrafast laser beam affect the slicing performance?

3. Compared with the non-diffractive ultrafast Bessel laser beam and femtosecond laser double pulsed assisted SiC slicing, what are the possible advantages of PLSD? Please clarify in detail.

Round 2

Reviewer 2 Report

The revised manuscript looks fine to me.

The article is acceptable for publication.

Reviewer 3 Report

All my comments have been addressed. The revised manuscript can be considered for acceptance by Micromachines.